# Formaldehyde Emissions from Wooden Toys: Comparison of Different Measurement Methods and Assessment of Exposure

**DOI:** 10.3390/ma14020262

**Published:** 2021-01-07

**Authors:** Morgane Even, Olaf Wilke, Sabine Kalus, Petra Schultes, Christoph Hutzler, Andreas Luch

**Affiliations:** 1German Federal Institute for Risk Assessment (BfR), Department of Chemical and Product Safety, Max-Dohrn-Strasse 8-10, 10589 Berlin, Germany; christoph.hutzler@bfr.bund.de (C.H.); andreas.luch@bfr.bund.de (A.L.); 2Bundesanstalt für Materialforschung und -prüfung (BAM), Division 4.2–Materials and Air Pollutants, Unter den Eichen 44-46, 12203 Berlin, Germany; olaf.wilke@bam.de (O.W.); sabine.kalus@bam.de (S.K.); 3Freie Universität Berlin, Department of Biology, Chemistry, Pharmacy, Institute of Pharmacy, Königin-Luise-Strasse 2-4, 14195 Berlin, Germany; 4Chemical and Veterinary Analytical Institute Münsterland-Emscher-Lippe (CVUA-MEL), Joseph-König-Str. 40, 48147 Münster, Germany; petra.schultes@cvua-mel.de

**Keywords:** formaldehyde, wooden toys, emission test chamber, flask method, EN 717-3, microchamber

## Abstract

Formaldehyde is considered as carcinogenic and is emitted from particleboards and plywood used in toy manufacturing. Currently, the flask method is frequently used in Europe for market surveillance purposes to assess formaldehyde release from toys, but its concordance to levels measured in emission test chambers is poor. Surveillance laboratories are unable to afford laborious and expensive emission chamber testing to comply with a new amendment of the European Toy Directive; they need an alternative method that can provide reliable results. Therefore, the application of miniaturised emission test chambers was tested. Comparisons between a 1 m^3^ emission test chamber and 44 mL microchambers with two particleboards over 28 days and between a 24 L desiccator chamber and the microchambers with three puzzle samples over 10 days resulted in a correlation coefficient r^2^ of 0.834 for formaldehyde at steady state. The correlation between the results obtained in microchambers vs. flask showed a high variability over 10 samples (r^2^: 0.145), thereby demonstrating the error-proneness of the flask method in comparison to methods carried out under ambient parameters. An exposure assessment was also performed for three toy puzzles: indoor formaldehyde concentrations caused by puzzles were not negligible (up to 8 µg/m^3^), especially when more conservative exposure scenarios were considered.

## 1. Introduction

Formaldehyde, the simplest aldehyde (HCHO), is colourless and detectable in the gas phase at ambient temperature. It is mainly used in the production of industrial resins, adhesives, and coatings. Since it was demonstrated to induce tumours in the nasopharynx of rodents [1], it has been classified as a carcinogen of category 1B since 2016 [2]. Formaldehyde scored highly as one of the top chemicals for both exposure and toxicity in Washington, USA [3], and in the European Union [4].

The German committee on indoor guideline values determined a guideline value of 100 µg/m^3^ based on toxicological data [5], which is in line with the WHO guideline [6]. An initial German survey in the years 1985–1986 revealed indoor formaldehyde concentrations of up to 309 µg/m^3^, with a mean concentration of 59 µg/m^3^ from 329 measurements [7]. In the following years, great efforts were made to reduce the formaldehyde sources and lower indoor air concentrations were measured, with a maximum of 68.9 µg/m^3^ during 2003–2006, for example [8]. A recent statistical review analysis from 2019 indicates that average concentrations of formaldehyde are within the range of 20–30 µg/m^3^ for European households under typical residential conditions [9].

Wood-based materials made of urea-formaldehyde resins may emit high formaldehyde concentrations [10,11]. They are mainly used as building materials or in the manufacturing of furniture, which caused 70% of formaldehyde indoor air concentrations in newly built timber-frame houses [12]. Urea-formaldehyde adhesives have poor water resistance: the presence of water causes hydrolysis and, consequently, the release of formaldehyde [13]. The European standard EN 717-1 suggests determining the release of formaldehyde from wood-based panels through the emission test chamber method [14]. The test chamber method is regarded as the method of choice for emission measurements as it mimics a real indoor environment (air exchange, temperature and humidity). Since 2017, the new standard method EN 16516 is in place in Europe: it describes emission testing with lower air change rate, higher relative humidity and higher chamber loading factor than EN 717-1 [15]. Since January 2020, the German national chemicals prohibition ordinance sets stricter requirements as EN 16516 must now be used instead of EN 717-1 to comply with the concentration limit of 0.1 ppm (corresponding to 124 µg/m^3^) for formaldehyde [16]. For the same chamber loading, EN 16516 leads to measured concentrations being a factor of 1.6 higher compared to EN 717-1 [10]. With a higher chamber loading of 1.8 m^2^/m^3^ instead of 1.0 m^2^/m^3^, a factor of 2 could be expected. According to EN 717-1, the air samples from test chamber measurements are analysed by photometry after reaction with acetylacetone or with liquid chromatography (HPLC) after derivatisation with 2,4-dinitrophenylhydrazine (DNPH), following ISO 16000-3 [17].

Toys made of wood-based panels may also emit formaldehyde. However, their origin and quality are not typically controlled in the same way as particleboards because they are usually directly imported from distant countries. The European Toy Safety Directive 2009/48/EC [18] specifies a general maximum level of 0.1% (1000 mg/kg_toy_) for carcinogenic compounds such as formaldehyde; however, this represents only a content limit and does not account for its emission behaviour. As formaldehyde is usually present in a chemically bound form and only emitted after hydrolysis, a content analysis for formaldehyde does not give any indications on the inhalation exposure assessment.

The so-called flask method is widely used by official control laboratories (OCLs) which are responsible for the toy market surveillance in the EU member states to measure formaldehyde emission of products [19]. It was developed by Roffael in the 1970s [20] and adapted into the European standard EN 717-3 [21]. The tested material is placed into the headspace of a 500 mL bottle filled with 50 mL water. After the incubation period of 3 h at 40 °C, the amount of formaldehyde dissolved in the water is determined by photometry. The method is still in use for wooden toys because of a lack of alternative methods, although it has been proven that the correlation to emission chamber testing is poor [22]. Moreover, different limits are used in the practice: EN 71-9 stipulates a maximum level of 80 mg/kg_toy_ if EN 717-3 is used (3 h experiment) [23], whereas the former German Federal Health Agency (BGA) recommended a limit of 110 mg/kg_toy_ for a 24 h flask experiment [24]. Using different materials, a study demonstrated that the values obtained by the flask method remained linear over time for at least 30 h [22], meaning that the two different limits are not comparable. The same study also suggested using an emission chamber test for more realistic results. There were several discussions at the subcommittees of analytics and toys related to the BfR’s committee for consumer products where German OCLs asked for advice and developments of reliable measurement methods for formaldehyde in wooden toys with respect to children’s safety [25].

In November 2019, a new European directive was adopted, amending 2009/48/EC for the purpose of specific limit values for chemicals used in certain toys [26]: here, in compliance with the German Chemicals Prohibition Ordinance [27], an emission limit of 0.1 ppm was stipulated for formaldehyde from resin-bonded material, starting from May 2021. In addition, the working group recommended emission testing by following EN 717-1 (i.e., a standardised method for wood-based panels) [14]. However, the OCLs will not be able to afford emission chamber testing for every toy and are therefore in need of an alternative method which provides reliable results. Smaller test chambers are cheaper, adapted to the typical size of toys and enable a higher sample capacity; their comparability to the standard chambers should be assessed considering the results obtained by the flask method.

Several studies have compared methods for determining formaldehyde emissions in the past. Firstly, the Field and Laboratory Emission Cell (FLEC) was compared to a standard 1 m^3^ emission chamber and provided good correlation [28]. Unfortunately, this method cannot be used for toys, which do in most cases do not have flat surfaces. In another study, most standard methods were compared and showed sample-dependent results [29]. This may have been influenced by the fact that test conditions also vary between different standards. Three environmental chambers of different sizes were also compared for formaldehyde emissions from carpets [30]. In this case, the test conditions (temperature, humidity, air change rate and loading factor) were kept constant but considerable differences in formaldehyde emissions could still be observed. These previous studies did not consider the use of microchambers (µ-CTE) which allow cheaper measurements of small products in replicates and already showed good correlation for VOC emissions from a polymeric material [31]. The µ-CTE is a device with six 44 mL (or four 114 mL) miniaturised emission test chambers where the temperature, humidity and air change rate are controlled: the air can be sampled at the chamber outlet [32]. To our knowledge, microchambers have so far never been compared to large and regular emission chambers in terms of formaldehyde emission testing. The so-called “Dynamic microchambers” (DMC) were used on particleboards by Hemmilä et al. (2018) [11] and compared with a 1 m^3^ emission test chamber and the perforator method (ISO 12460-5 [33]). However, DMC have a much higher volume (44 L) than the microchambers used in this study and are therefore linked with higher operating costs. Another micro-scaled chamber (1 L) that allowed process automation was tested for formaldehyde emission. However, no correlation with standard emission chambers could be demonstrated [34].

A standard cost-effective routine method usable for formaldehyde emission testing of toys and other consumer products in OCLs still needs to be established. Thus, we tested the comparability of formaldehyde emissions from wooden products in emission test chambers of different sizes and with the flask method: we demonstrated that microchambers can be used as a good alternative to the existing methods. Finally, we estimated the corresponding inhalative exposure against formaldehyde from wooden toys and showed that it was not negligible.

## 2. Materials and Methods

### 2.1. Samples

An overview of the samples used is given in Table 1, the exact dimensions are provided in Appendix A. Two particleboards were initially studied. They were bought from a local do-it-yourself store and had already shown relatively high formaldehyde emissions during previous tests two years earlier [10]. Eight different wooden toys were also investigated. They were bought in local stores and had shown (except for Sample #9) flask method values (40 °C, 24 h) beyond the limit of 110 mg/kg_toy_ recommended by the former German Federal Health Agency (BGA) [24] during market surveillance (see Appendix A for the exact values). Their country of origin was always China if it could be identified, meaning that the initial wood-based materials had not necessarily been controlled according to European standards [14,15]. Until usage, the samples were kept at room temperature in their original packaging or covered with aluminium foil. Pictures of the samples are provided in Appendix A.

### 2.2. Emission Test Chambers

Three different types of emission test chambers (1 m^3^, 24 L, and 44 mL) were used for emission testing, along with a clean air supply system. The 1 m^3^ chamber was the standard VOC emission test chamber model from Heraeus-Vötsch Industrietechnik (Balingen-Frommern, Germany) with an inner chamber made of electro-polished stainless steel and a ventilator to ensure homogeneous air distribution. The 24 L chambers were desiccators made of glass and equipped with a ventilator from the BAM (Bundesanstalt für Materialforschung und -prüfung, Berlin, Germany). They were used instead of the 1 m^3^ test chambers as standard chambers for the wooden puzzles because some samples were too small to obtain meaningful concentrations in the bigger chambers. The 44 mL chambers were part of a micro-chamber/thermal extractor device (μCTE^®^) produced by Markes (Llantrisant, UK).

The edges of the particleboard pieces (two plates of 0.5 m × 0.5 m and 0.43 m × 0.5 m in the 1 m^3^ chamber and 1 piece of 2 cm × 4 cm in the microchambers) were covered with an emission-free aluminium-coated tape according to EN 717-1 [14]. The ratio between the open edge and the total surface was adjusted to 1.5 m/m^2^. Some toy samples had to be cut with a saw to fit into the microchambers (#5, #6, #8 and #10). In this case, the freshly cut edges were covered completely with tape; indeed, the non-geometrical form of the toy makes it difficult to cover a defined ratio of the edges.

The two particleboards were placed upright in the 1 m^3^ chamber. The puzzle and toy pieces were placed on metal carriers in the desiccators and on small plastic carriers in the microchamber if air would not otherwise circulate under the sample. Pictures of chamber loading are presented in Appendix A. Replicates were used for the microchambers: two or three chambers were always loaded with similar pieces of the same sample.

The systems were set to a temperature of 23 ± 1 °C and 50 ± 5% relative humidity. The microchambers were operated at a flow of 23.1–29.3 mL/min, while the desiccators were operated with 1.80 and 1.88 L/min. Similar to our previous work [31], the air change rate in the 1 m^3^ chamber was adapted to the chamber loading to obtain a similar area-specific airflow rate (ratio of air change rate to loading) as applied for the microchamber, resulting in a flow of 14.5 L/min. Evidently, this represents a crucial parameter for such studies [35] and should be kept as constant as possible. Despite the maximum possible loading of the desiccator (all the puzzle pieces with the exception of the one placed in the microchambers), the area-specific air flow for chamber comparison was lower in the microchamber but still in the same order of magnitude. The area-specific values for air flow used during chamber comparisons are summarised in Table 2. To compensate the discrepancies, the results of method comparisons are presented as surface area specific.

### 2.3. Air Sampling and Analysis of Air Samples

Air sampling was performed using DNPH cartridges (Supelco, St. Louis, MO, USA). The DNPH method [17] was preferred to the photometry method [14] for sample analysis because it was already widely used and validated in our laboratory. Active sampling was carried out for the 1 m^3^ chamber and desiccators following ISO 16000-3 [17] using an air check 3000 sample pump (SKC Ltd., Dorset, UK) at 1 L/min for 30 min. Two samples were collected simultaneously for each time point in the 1 m^3^ chamber: a self-designed sampling pump was used for the second sample. For the microchambers, the sampling lasted 20 h at the outlet to allow a sampling volume of around 30 l. Several samples were taken before the actual measurements started to control for blank values of the chambers and the DNPH cartridges. Air samples were regularly collected over 28 or 10 days after loading of the chambers.

The cartridges were refrigerated before and after sampling and eluted with 2 mL acetonitrile within two weeks after sampling. The solutions were analysed using HPLC (HP1100 from Hewlett-Packard, Waldbronn, Germany) in accordance with ISO 16000-3 [17]. An UltraSep ES ALD column (125 mm × 2.0 mm) and a pre-column (10 mm × 2 mm) from SepServ (Berlin, Germany) were used. The gradient of acetonitrile to water + 6% tetrahydrofuran varied between 30% and 83% (30% hold for 5 min, to 32% in 5 min and hold for 20 min, to 83% in 25 min). The mobile phase flow was 0.5–0.6 mL/min and the Diode Array Detector was used at 365 nm. Formaldehyde was quantified via external calibration with a commercial solution of its derivative from Sigma-Aldrich (Darmstadt, Germany) with a maximum concentration of 50 ng/µL. Samples were diluted if they did not fit into the calibration range. Data was processed using the OpenLab Data Analysis A.01.02 software from Agilent (Waldbronn, Germany). The results are provided as area-specific emission rates (SER_A_), weight-specific emission rates (SER_W_) or indoor air concentrations (C_indoor_):(1)$${\mathrm{SER}}_{A}=\frac{C_{\mathrm{CH}}{*V}_{\mathrm{CH}}{*n}_{\mathrm{CH}}}{A}$$
where SER_A_ is the area-specific emission rate (mg/h·m^2^); C_CH_ is the chamber concentration (mg/m^3^); V_CH_ is the chamber volume (m^3^); n_CH_ is the chamber air change rate (/h); and A is the sample surface area (m^2^).
(2)$${\mathrm{SER}}_{W}=\frac{C_{\mathrm{CH}}{*V}_{\mathrm{CH}}{*n}_{\mathrm{CH}}}{m}$$
where SER_W_ is the weight-specific emission rate (mg/h·g); and m is the sample weight (g).
(3)$$C_{\mathrm{indoor}}=\frac{{\mathrm{SER}}_{A}*A}{V_{\mathrm{room}}{*\;n}_{\mathrm{room}}}=C_{\mathrm{CH}}*\frac{V_{\mathrm{CH}}{*\;n}_{\mathrm{CH}}}{V_{\mathrm{room}}{*\;n}_{\mathrm{room}}}$$
where C_indoor_ is the indoor air concentration (mg/m^3^); V_room_ is the room volume (30 m^3^ [15]); and n_room_ is the room air change rate (0.5/h [15]).

Surface areas of the samples were determined by approximating their shape to geometrical forms (e.g., ellipse and triangle, see Appendix A) if they were not already geometrical. For Sample #1, #2, #5, #7 and #8, all surface areas were determined. For the other samples, only the surface areas of the pieces placed in the microchambers were determined; the last approximation (3.4) with the whole sample surface area was done with the mean surface area of Sample #5, #7 and #8.

When two chambers were compared, an offset was calculated:(4)$$\mathrm{Offset}=\;\frac{\left({\mathrm{Highest}\;\mathrm{SER}}_{A}-{\mathrm{Lowest}\;\mathrm{SER}}_{A}\right)}{{\mathrm{Lowest}\;\mathrm{SER}}_{A}}\;\left(\%\right)$$

The use of the offset allows a direct comparison of the differences between emission test chambers for different samples.

The linearity of the correlation between SER_A_ at steady state in different emission test chambers was investigated. The coefficient of determination (R^2^) and the *p*-values were considered for statistical analysis of the linear regressions. *P*-values were computed with the mean of each data point and were considered statistically significant when < 0.05 and highly statistically significant when < 0.001.

### 2.4. Flask Method

The flask method was carried out the same way as it is done by toy market surveillance [24]: in accordance with EN 717-3 [21] at 40 °C but for 24 h. The results are given in mg formaldehyde released per kg toy (mg/kg_toy_). The linearity of the correlation between the flask method values and the emission rates after 10 or 11 days in the microchamber was investigated.

For Samples #3–#7, the test was conducted again after microchamber testing to study the influence of emission testing on the flask method values. 

Except for the samples (particleboards and wooden toys), which are possibly only purchasable for a restricted time frame, and the desiccators which were self-made, all the materials and equipment used in this study are available commercially.

## 3. Results and Discussion

### 3.1. Chamber Comparison Using Particleboards

The 1 m^3^ chamber and the microchamber were first loaded with pieces from the same particleboards and air samples were collected regularly over 28 days. Area-specific emission rates are depicted in Figure 1.

Firstly, it was observed that both chamber types led to similar emission profiles for formaldehyde: area-specific emission rates were relatively constant over 28 days, probably due to a year-long storage under chamber climate similar conditions. Emission rates were always used for test chamber comparisons because it is directly related to the indoor air concentration (see Equation (1)) but normalised to the area-specific air flow rate. Secondly, a relatively stable offset was observed between both chamber types: the emission levels measured in the microchamber were in mean about +27% and +28% (offset calculated according to Equation (4)) compared to those of the 1 m^3^ chamber. A possible reason for the observed discrepancies could be the covering of the open edges with a ratio to the total surface of 1.5 m/m^2^ as stipulated by EN 717-1 [14]: it represented 2.4 mm of open edges for the 2 cm × 4 cm pieces placed in the microchambers, which is difficult to accurately achieve using tape. Differences in air velocities at the sample’s surface could also explain this deviation between both chambers. However, it is not possible to measure the air velocity in the microchambers.

These data still indicate that a good correlation between both chamber sizes was observed under the selected conditions. Thus, small chamber sizes might be a promising alternative for cost-effective emission measurement of formaldehyde from particleboards.

It has been demonstrated in previous work that the flow circulation in the microchamber is heterogeneous [32]. The height of the sample could disturb the air flow and thus would have an influence on the emission. This is of particular importance in the case of specimens that represent one-third of the chamber volume. For this reason, formaldehyde emission was analysed for different positions of both Samples #1 and #2 in the microchamber. The results are provided in Figure 2.

This experiment revealed that the position of the sample in the microchamber is only of low importance: irrespective of the exact position, the area-specific emission rate was similar (same position repeated or new position tested). This is an important result as it means that the exact position of the sample in the emission chamber would not be a crucial parameter in market control experiments. Furthermore, Figure 1 and Figure 2 both show that an emission rate equilibrium is reached very quickly. For this reason and to allow fast and efficient investigations the following experiments were limited to 10 days.

### 3.2. Chamber Comparison for Toy Samples

A similar experiment to the one presented in Section 3.1 was conducted using wooden puzzles. Most puzzle pieces fit into the microchamber or easily fit when cut, and the cutting edge was covered by aluminium tape. Puzzle or play set pieces were chosen for this comparison; one piece was loaded into the microchamber while all other available pieces (8–18) of the same sample were loaded into a desiccator chamber, resulting in area-specific air flow rates that were slightly higher in the desiccator (see Table 2) compared to the microchamber. Three samples were studied over 10 days. The formaldehyde emission results are shown in Figure 3.

Formaldehyde concentrations were found to be fairly constant after seven days. Over the course of 10 days, the area-specific emission rates were similar in both chambers for all three samples. For Samples #5 and #7, there was no significant offset between both chamber results in contrast to Figure 1. For Sample #8, the average offset between microchamber and desiccator results was +53%, slightly higher than for Samples #1 and #2. For Sample #8, the area-specific air flow rate was only 1.9 times higher in the desiccator than in the microchamber while the ratio between both chambers was 2.3 and 2.6 for Samples #5 and #7, respectively (see Table 1). This finding may contribute to the fact that Sample #8 behaved similarly to Samples #1 and #2 (for which the area-specific air flow was constant between chambers). Moreover, the shape of the pieces from Sample #8 (play set) were thicker and approached the shape of Samples #1 and #2 more than the puzzle pieces from Samples #5 and #7. This may lead to differences in air velocities at sample surface. Additionally, for Samples #5 and #7, a decrease of the formaldehyde emission rate is observed over the first few days. Such a decrease was not observed for Sample #8 or for the particleboards in Figure 1. These differences can be due to more consistent conditions during storage or to the fact that Samples #5 and #7 were coloured with stickers while Sample #8 was painted (see Appendix A). Stickers could emit high formaldehyde concentrations during the first hours. The decrease was also observed for Sample #6 (see Appendix A). Hemmilä et al. (2018) also observed that the formaldehyde profile before steady state depended on the sample: no linear correlation was found between DMC results at Days 1 and 7 for different samples [11].

The results obtained from the microchamber and the desiccator experiments are similar, especially after 10 days. Again, the microchamber seems to provide reliable and comparable formaldehyde emission results. When using the microchamber, temperature and humidity are regulated according to EN 717-1 [14], but the chamber loading factor and the air change rate are higher to achieve similar area-specific air flows (see Table 2). In consequence, this leads to the conclusion that the standard method cannot be applied word by word. Furthermore, it will be difficult in practice to comply with the standard EN 717-1 [14] as it requires that sample edges should be partly covered by a special ratio and that a certain loading factor should be used. This is much more complicated for toy samples than for large and rectangular wood-based panels. A practical suggestion would be to completely cover cut edges with tape before sample loading.

The linear correlation between the emission rates measured at steady state in the emission test chambers and the microchambers is shown in Figure 4. For Samples #1, #2 and #8, the whole measurement period was considered as steady state. For Samples #5 and #7, only the measurements at Days 7 and 10 were considered.

A good correlation was observed between the microchamber and the test chamber (1 m^3^ or desiccator) results, with a R^2^ of 0.834 and a significant *p*-value of 0.0304 (<0.05). A slope of 0.9944 indicates a good matching between chambers of different sizes.

### 3.3. Feasibility of the Flask Method

For all investigated samples, microchamber and flask experiments were carried out using two pieces of the same sample that were as similar as possible. The correlation between the flask method value and the emission rates at Day 10 (or Day 11 for Samples #1 and #2) is presented in Figure 5.

In Figure 5a, both the R^2^ (0.145) and the *p*-value (0.2775 > 0.05) indicate a poor correlation between flask method values and area-specific emission rates (SER_A_) of the samples. If Samples #3, #4 and #8 are removed, a R^2^ of 0.956 is obtained, indicating a good linear correlation with a highly significant *p*-value (0.00014 < 0.001). As observed by Hemmilä et al. (2018) with the perforator method, the correlation between the emission chamber and the flask method result is consistent only for a selection of samples [11]. The results seem to depend on the geometry of the toy. Samples #3 and #8 were the thickest samples (1.4 and 1.8 cm) and yielded the lowest ratios of the flask method value to SER_A_, while Sample #4, as the thinnest sample (0.3 cm), was found to result in the highest ratio. Hemmilä et al. (2018) also observed a differentiation between samples with varying thickness [11].

SER_A_ is a common unit for material emission measurements but as the flask method value is based on the weight of the toy, a correlation with the weight-specific emission rate (SER_W_) was also considered and is presented in Figure 5b. In this case, a better correlation is obtained between the flask method values and the weight-specific emission rate, with a statistically significant correlation (*p*-value: 0.02876 < 0.05) but a poor R^2^ (0.470). If only Sample #4 is removed, a R^2^ of 0.845 and a highly statistically significant *p*-value of 0.0005 (<0.001) are obtained. The correlation between the microchamber and the test chamber (1 m^3^ or desiccator) results is better (R^2^: 0.834, see Figure 4). An interesting observation was that Sample #3 seemed to be made of massive wood and still emitted as much formaldehyde (1.4 mg/h·m^2^ at Day 10) as the other samples. Such results have also been observed in the past [36] and may be attributed to the lacquer. The area-specific emission rate curves from the toy samples over 10 days in the microchamber are provided in Appendix A.

Overall, the flask method is not a good way to predict the emission measurements performed under realistic environmental conditions for varying toys. This mirrors the evidence obtained by other studies [22,29]. The flask method also has the significant disadvantage that the sides cannot be covered permanently (the humidity is too high for the tape) if the sample needs to be cut.

Additionally, the influence of the time point of the flask method test on the results was studied. The results presented above consider the flask experiment conducted with samples similar to those used in the microchamber. The flask method was carried out again for some samples that had undergone the microchamber experiment (Samples #3–#7) and the observed values are shown in Appendix A together with those of the similar samples. Generally, no significant difference was observed between both values (margins of error overlap). Significant differences were only observed for Sample #4: this would enhance the correlation with the microchamber results, which will however stay poor (R^2^: 0.517). Samples #5 and #6 resulted in similar values in both scenarios: it seemed to be of minor significance whether they had an open edge (following the chamber experiment).

### 3.4. Exposure Assessment from Whole Toy Samples

As shown in Section 3.2, puzzle or play set pieces were studied in a desiccator for comparison with the formaldehyde emission in the microchamber in which they could often fit without further sample preparation. The puzzle plates can also emit formaldehyde, but they are not investigated by OCLs because they also do not fit in the flask. It may be possible that the plate is the part with the greatest emissions. However, this is not necessarily considered for market surveillance or exposure assessment purposes. An evaluation of the contribution of both sample parts is presented: puzzle plates for each of the three samples (Samples #5, #7 and #8) were also studied in a desiccator for 10 days. The results of the piece-specific emission rates, normalised to one plate or to the number of associated pieces, are presented in Figure 6.

For Samples #5 and #7, the plate was emitting higher amounts of formaldehyde compared to the puzzle pieces. However, for play set Sample #8, the pieces were emitting more formaldehyde compared to the plate. These differences are likely due to the different geometries of the samples: the play set plate (682 cm^2^) is smaller compared to the puzzle plates (1426 and 1456 cm^2^), whereas the sum of the piece surface areas was similar between samples (474–518 cm^2^). The results in area-specific emission rates are similar for the pieces and the plate for Samples #5 and #7. For Sample #8, pieces emitted much more formaldehyde per surface unit than the plate: this is probably partly due to the fact that they were thicker (typically 1.8 cm) than the plate (0.5 cm). This shows that every part of a toy should be investigated when exposure needs to be assessed.

Furthermore, an exposure assessment of formaldehyde was carried out using the results of the desiccator experiments. The influence of a whole puzzle set (plate and corresponding number of pieces, as a consumer would buy it; see number of pieces in Table 1) on the formaldehyde room concentration has been considered. Evaluating indoor air concentrations allows a direct comparison with the indoor air guideline and therefore a reliable risk assessment [37]. They were calculated as shown in Equation (3), and the results are displayed in Figure 7.

In the indoor air scenario (Figure 7), the influence of one sample on the formaldehyde concentration is considered in a 30 m^3^ room with an air change rate of 0.5/h [15]. The assessment reveals that whole puzzle samples could induce indoor air concentrations of formaldehyde of up to 12 µg/m^3^ on the first day and 5 µg/m^3^ after 10 days. In comparison, the indoor air guideline value [5] is 100 µg/m^3^. However, the children may play in very close proximity to the product in a poorly ventilated space with a concentration gradient. The concentration in the child’s breathing zone will then be higher than the average room concentration. Moreover, there may also be other formaldehyde sources in the indoor air environment, meaning that the contribution of such products cannot be considered negligible. The BfR stated in 2007 that emissions from toys should only contribute to 10% of the overall indoor formaldehyde guideline concentration [38]. As an example, Samples #5, #7 and #8 represent, respectively, 4.8%, 4.6% and 1.7% of the 100 µg/m^3^ guideline at steady state. An exceedance is easily possible if a room containing several toys and other formaldehyde sources such as furniture or building products is considered. It should also be considered that an increased temperature and/or humidity can enhance formaldehyde emissions drastically [39] and therefore cause even higher exposures. Lower air change rates indoors can also lead to higher VOC concentrations. A reduction of the formaldehyde emission limit is currently under discussion in Europe. Lower limits are already in effect for certain types of samples in the USA [40] and in Japan [41].

A similar exposure assessment for formaldehyde could be carried out directly using the area-specific emission rates from microchamber studies as they correlated with those obtained from bigger emission test chambers. The results from this approximation are shown in Table 3. This would lead to slightly higher concentrations for Samples #3 and #6 than for the previous considerations.

The difference between the approximated formaldehyde concentrations of the desiccator and the microchamber for Sample #8 is due to the plate, which emitted less formaldehyde than the pieces and was not considered in the microchamber approximation. This underlines the fact that representative pieces (e.g., a cut piece of the plate with covered edges) should be analysed if using the microchamber. With this consideration, the microchamber seems to be an appropriate method for market surveillance. The approximation carried out with only pieces of Sample #5 and #7 are close to the values obtained with the desiccator.

## 4. Summary and Conclusions

Formaldehyde emission is a key aspect when ensuring wooden toy safety. The emission test chamber method described in EN 717-1 [14] is not practicably and economically feasible for measurement purposes of toys. There is an urgent need for a standardised measurement method which demonstrates a good correlation to the existing emission test chamber methods whilst being more cost-effective. A possible method was investigated in our present study: the capacity of miniaturised emission test chambers (44 mL) to facilitate the surveillance of formaldehyde emissions from wooden toys was evaluated. It was shown that the emission results obtained (area-specific emission rates) were comparable to those from bigger chambers for both particleboards and wooden toys. In contrast, the currently used flask method showed a bad correlation with emission test chamber results. Its further use for market control of wooden toys is highly questionable. The main drawback in suggesting large-volume emission test chambers for toy market surveillance are higher costs and low sample capacity. Therefore, microchambers represent an affordable alternative for reliable market surveillance by the OCLs. They show a statistically significant correlation with bigger chambers (overtime and at steady state with *p* < 0.05 and R^2^: 0.834), but further investigations with a larger number of samples and a validation using a homogeneous material are suggested to support these findings before standardisation.

As the sample may be heterogeneous, it is necessary to analyse representative pieces of the toy. Moreover, the standard EN 717-1 [14] is not directly suited to microchamber testing of toys. The air change rate will be higher than 1/h and the toy edges cannot be covered with a specific ratio. In addition, it is impossible to use a defined chamber loading factor for wooden toys due to the variable shapes. In the most recent standard EN 16516, different loading factors are stipulated for different sample types [15]. The new amendment of the toy safety directive [26] could indicate either an area-specific emission limit or a concentration limit per toy piece instead of following EN 717-1 with a defined chamber loading. Additionally, other analytical techniques, such as photometry [14], could be considered for air sample analysis to further reduce measurement costs.

An exposure assessment led to notable indoor air concentration values, indicating that formaldehyde emissions from a single wooden play set could represent up to 8% of the WHO formaldehyde guideline for indoor air. These products should therefore be considered as important emission sources, especially if numerous wooden toys are kept in smaller rooms or if children play with such toys and keep them in close proximity to their breathing zone.

## Figures and Tables

**Figure 1 materials-14-00262-f001:**
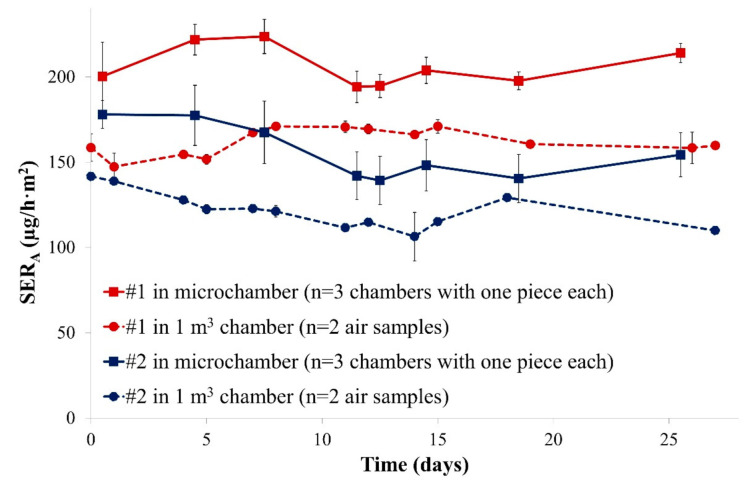
Emission profiles of formaldehyde from the two particleboards in two different emission chambers over four weeks (SER_A_, area-specific emission rate). Error bars represent SD (standard deviation).

**Figure 2 materials-14-00262-f002:**
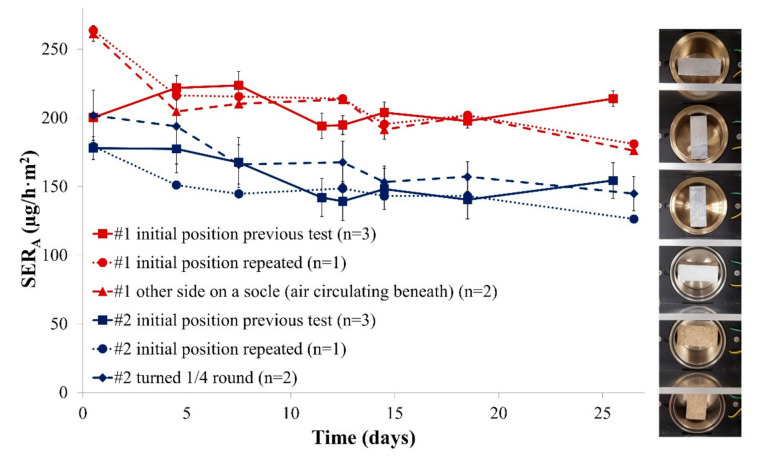
(**Left**) Emission profiles of formaldehyde released from two particleboards for different sample positions in the microchamber (SER_A_, area-specific emission rate); and (**Right**) picture of the different testing positions. Error bars represent SD.

**Figure 3 materials-14-00262-f003:**
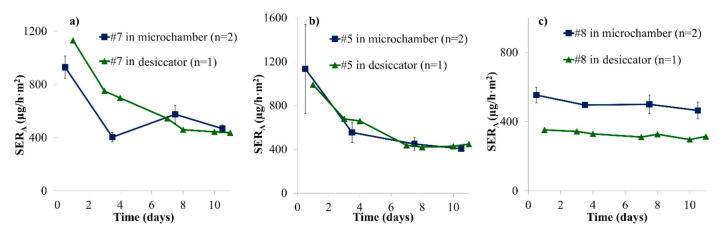
Emission profiles of formaldehyde from puzzle pieces in two different emission chambers over 10 days (SER_A_, area-specific emission rate). (**a**) Sample #7; (**b**) Sample #5; (**c**) Sample #8. Error bars represent SD.

**Figure 4 materials-14-00262-f004:**
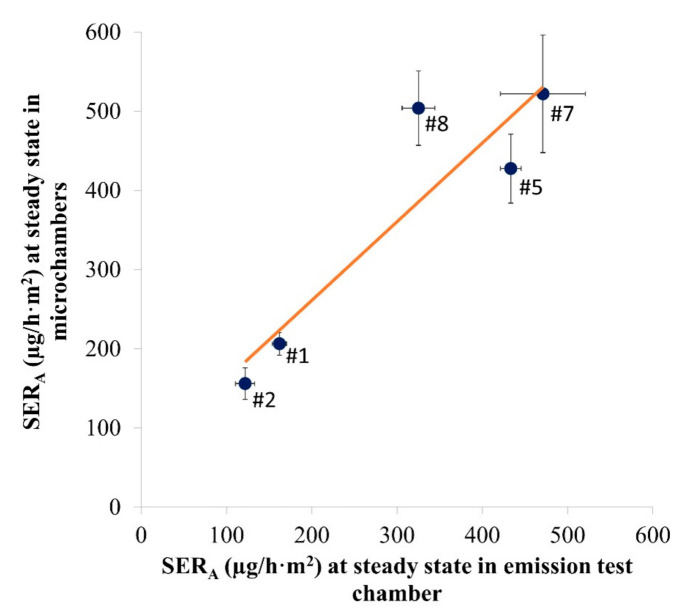
Study of the linearity between the area-specific emission rate (SER_A_ in µg/h·m^2^) in the emission test chamber and in the microchambers for Samples #1, #2, #5, #7 and #8. R^2^: 0.834; *p*-value: 0.0304 (*n* = 4–24, depending on the number of air samples during steady state). y = 0.9944x + 62.113. Error bars represent SD.

**Figure 5 materials-14-00262-f005:**
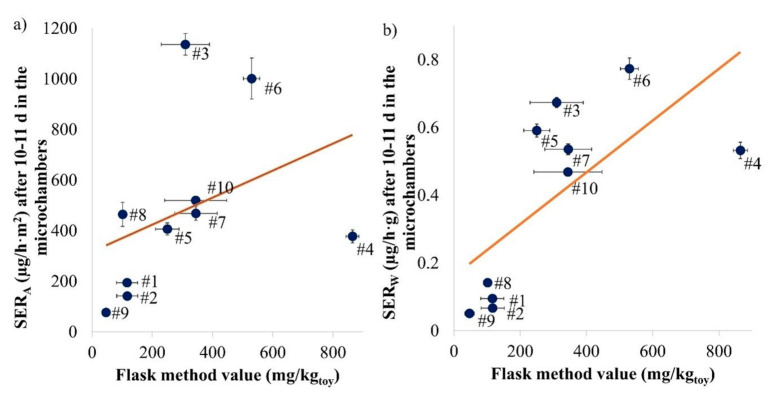
Study of the linear correlation between the flask method values and the area-specific ((**a**) R^2^: 0.145; *p*-value: 0.27775) or weight-specific ((**b**) R^2^: 0.470; *p*-value: 0.02876) emission rates after 10 or 11 days in the microchamber for each sample investigated. Error bars represent SD (*n* = 2 for microchamber samples, for flask method see Appendix A).

**Figure 6 materials-14-00262-f006:**
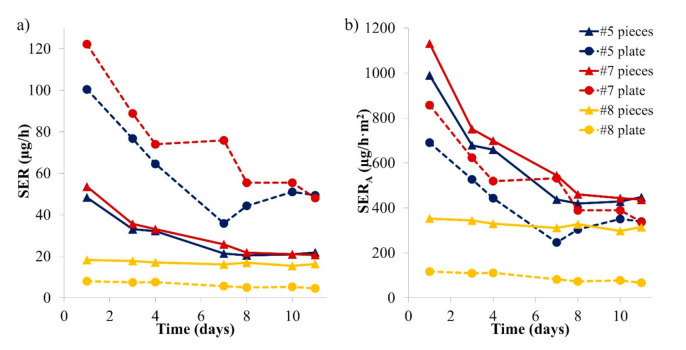
Emission rates of formaldehyde from the plate and the pieces of three puzzles: SER (sample-specific emission rate) (**a**); and SER_A_ (area-specific emission rate (**b**)). The results were obtained with the desiccator method (*n* = 1).

**Figure 7 materials-14-00262-f007:**
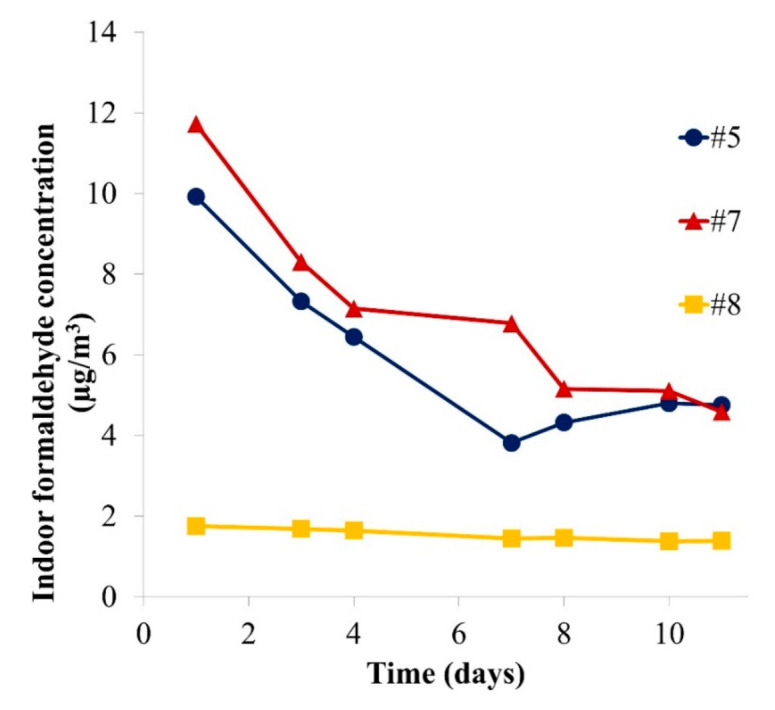
Calculated formaldehyde concentration for a 30 m^3^ room resulting from each puzzle set sample (plate and pieces) (*n* = 1).

**Table 1 materials-14-00262-t001:** Overview and dimensions of the samples studied; ^¥^: the puzzle pieces were cut to fit in the microchambers, open edges were partly (#1 and #2) or entirely (#5, #6, #8 and #10) covered.

**No.**	**Description**	**Origin**	**Sample Surface Area (cm^2^)**			
			**Microchambers**	**1 m^3^ Chamber**		
#1	Particleboard	E.U.	16.0 ^¥^	9300		
#2	Particleboard	E.U.				
#3	Block set	China	34.4	**Desiccator**		**Number of Pieces Per Set**
#4	Hammer and nail set	unknown	28.9	**Plate**	**Pieces**	
#5	Puzzle birds	China	19.7 ^¥^	1456	651	12
#6	Puzzle fish	China	12.6 ^¥^			
#7	Puzzle shapes	unknown	24.0	1426	711	12
#8	Play set meal	China	23.9 ^¥^	1475	828	5
#9	Puzzle numbers	unknown	29.0			
#10	Plug set garden	unknown	16.9 ^¥^			

**Table 2 materials-14-00262-t002:** Sample area-specific air flow (m^3^/m^2^·h) values for chamber comparisons (n.u., not used; −, range due to flow fluctuation).

No.	Microchamber	Desiccator	1 m^3^ Chamber
#1	0.97−1.06	n.u.	0.94
#2	0.95−1.03	n.u.	0.94
#5	0.72−0.79	1.73	n.u.
#7	0.61−0.63	1.59	n.u.
#8	0.68−0.74	1.36	n.u.

**Table 3 materials-14-00262-t003:** Room concentration at Day 10 or 11 approximated from microchamber measurements.

	#3	#4	#5	#6	#7	#8	#9	#10
Approximated C_room_ from microchamber (µg/m^3^) (*n* = 2)	8.27	2.75	3.29	7.29	3.71	3.71	0.56	3.78
	±0.31	±0.18	±0.18	±0.59	±0.20	±0.35	±0.01	±0.12
*C_room_ from desiccator (*µg/m^3^*)*			*4.78*		*4.84*	*1.71*		
Percentage of the WHO guideline	8.3%	2.7%	3.3%	7.3%	3.7%	3.5%	0.6%	3.8%

## Data Availability

The data presented in this study are available on request from the corresponding author. The data are not publicly available due to the absence of such requirements at the time the project was carried out.

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
