# Peer review of "Formaldehyde Emissions from Wooden Toys: Comparison of Different Measurement Methods and Assessment of Exposure"

_materials, 2021, doi:10.3390/ma14020262_

Round 1

Reviewer 1 Report

The paper presents an interesting topic but, due to the missing of details related to the samples characteristics (dimensions, geometry, thickness of every sample, introduced in a table) it is very hard to compare the results.

A more systematic presentation of the experiments and data will improve the understanding of novelty of this study and of the possibilities of application. 

Also a better presentation of the advantages, disadvantages and suitability of use for each method will be of interest.

Author Response

The authors would like to thank the reviewer for the constructive comments that helped us to improve the quality of our manuscript. Please find below our point by point responses to all comments:

The paper presents an interesting topic but, due to the missing of details related to the samples characteristics (dimensions, geometry, thickness of every sample, introduced in a table) it is very hard to compare the results.

Response of the authors: We provided more information on sample characteristics in Table 1, Figure S1 and Table S1.

A more systematic presentation of the experiments and data will improve the understanding of novelty of this study and of the possibilities of application.

Response of the authors: We provide more details in the experimental part and present the results in more details, together with a more extensive discussion in order to make the manuscript easier to read and to understand. The main novelty is clearly the use of the microchambers and their comparison with bigger emission test chambers and the flask method. The application we focused on is the toy market surveillance, because we assume this topic suffers on a lack of appropriate suitable analytical methods despite its very high socio-political and economic relevance. But we are confident that the results may be interesting in other fields as well.

Also a better presentation of the advantages, disadvantages and suitability of use for each method will be of interest.

Response of the authors: A presentation of the advantages, disadvantages and suitability for the use of each method was added in the conclusion (Line 455-460 in unmarked version).

Reviewer 2 Report

The objective of this manuscript was to apply miniaturised emission test chambers and compare them with standardised tests to determine the formaldehyde emission from wooden toys.

The paper is innovative, but the strategy to compare the methods is somehow unusual. Normally, it is established a correlation between the formaldehyde concentration obtained by the two methods instead of comparing the steady-state flow.

The state of the art section does not present the problematic of the formaldehyde emission from toys and the existing methods to determine it, in a clear way. This section starts speaking about the limit of the indoor formaldehyde concentration of formaldehyde and then the emissions from wood-based panels is approached. The materials and regulations related to toys should be approached first and then the methods used to determine formaldehyde emissions from wood-based panels, one of the material that could be used in toys. The recent strict regulations of formaldehyde emission, as CARBII legislation or F**** and also the latest German legislation from Federal Ministry for the Environment, Nature Conservation and Nuclear Safety of Germany made to their testing method, that effectively lowers the formaldehyde emission levels of wood-based panels in Germany from the European emission level of 0.1 ppm (E1, EN 717-1) to 0.05 ppm were not treated. The standard method EN 16516 indicated in this legislation is only mentioned in the conclusions. The example of emissions from carpets is somehow out of scope.

The sentence “Wood-based materials made of urea-formaldehyde resins are among the products that are responsible for the highest indoor formaldehyde emissions” is not really true, nowadays. The reference is from 1999. Nowadays, wood-based panels are produced with formaldehyde levels that complains strict regulations, even at solid wood level.

The sentence “Often, the analysis of air samples is carried out after derivatisation with 2,4-dinitrophenylhydrazine (DNPH), which is provided in cartridges, and successive elution and liquid chromatography following ISO 16000-3” should be better contextualized, because this procedure is the analysis of formaldehyde in the air. It should be mentioned that normally the analysis of formaldehyde in the chamber method is carried using the photometric method. Formaldehyde is absorbed in water and the water is analysed photometrically using the acetylacetone method. Moreover, for DNPH the system is expensive (cartridges, solvents and columns) and the formaldehyde in the acetonitrile reagent should be checked on a regular basis. It is mentioned in the conclusions that “Additionally, other analytical techniques, such as photometry could be considered for air sample analysis to further reduce measurement costs”, but the photometric method is not addressed.

I disagree that the flask method is widely used by official control laboratories (OCLs) in the EU member states to measure formaldehyde emission of products. In fact, the perforator and gas analysis methods are nowadays used by wood-based panels producers as quick methods for production control because the correlation to emission chamber testing is generally good. The accuracy and correlation of the flask method to the chamber methods are difficult to achieve due to small sample sizes, no air exchange, and high relative humidity (see DOI: 10.1039/c7ra06598a).

The observation “To our knowledge, microchambers have so far never been compared to large and regular emission chambers in terms of formaldehyde emission testing” is not truth. Please see the paper https://doi.org/10.1155/2018/4582383.

In materials and methods, all the samples should be clearly described or a photo should be presented. The estimated area is not presented. A small description of the Micro-Chamber/Thermal Extractor™ (μ-CTE™) should be also included.

The dimensions of the particleboard panels for used in chamber method are too small. The units are probably not correct. In EN 717-1, the dimensions of test pieces are normally 0,5 m x 0,5 m x thickness, which should correspond to a total loading rate of 1 m²/m³. In case of dessicator method, the test pieces should have 150 mm x 50 mm and the total surface area of 1800 cm2 including ends, sides and faces.

The results are not always well explained. Why the results of the evolution SER along the time from puzzle pieces attained nearly a steady-state condition and this evolution is very different in case of particleboard samples?

In Figure 6, it is presented the calculated formaldehyde concentration for a 30 m³ room resulting from each puzzle sample Please explain why in line 360, it is mentioned “The sum of the puzzle plate and the number of associated pieces has been considered”.

Considering this figure, the contribution of each puzzle piece for the indoor formaldehyde concentration in air for a 30 m³ room is very high in comparison with the indoor air guideline value (100 μg/m3).

Considering that the room (30 m3) is a CSTR (continuous stirred-tank reactor) with a flow of 15 m3/h, in which a formaldehyde source with 0.4 mg/(m2h), we will have a steady state concentration of C=0.4*A/15=0.026*A mg/m3. Given a concentration of more or less 0.004 mg/m3 in the room (see Fig. 6), the total surface of the sample should be around 0.15 m2, which is extremely large compared to a puzzle piece. Moreover, to exceed the WHO guideline, it will be necessary to have 3.75 m2 which corresponds to 1/3 of the total floor surface area of the room.

Some minor revisions are also indicated in the manuscript.

So, I think that the paper should be accepted with major revisions.

Author Response

The authors would like to thank the reviewer for the constructive and very detailed comments that helped us to improve the quality of our manuscript. Please find below our point by point responses to all comments:

The objective of this manuscript was to apply miniaturised emission test chambers and compare them with standardised tests to determine the formaldehyde emission from wooden toys. The paper is innovative, but the strategy to compare the methods is somehow unusual. Normally, it is established a correlation between the formaldehyde concentration obtained by the two methods instead of comparing the steady-state flow.

Response of the authors: It is unclear to the authors what is meant by comparing the steady state flows.  We assume that this comment is referred to the steady state emission rates instead of steady state flow rates. Since the results of the flask method are not analysed from the gas phase, they are not classical emission rates. We compared the flask method values with the emission rates at steady state (10 or 11 d) obtained with the microchambers. For the comparison between different emission chamber types with different sizes, we chose to follow the emission profiles overtime to better understand the emission processes and to have an experimental criterion whether the steady state emission rate is achieved after a certain time or not. For us it was not clear if steady state would be achieved at the same time point for the different chamber types. Indeed, for such products which may be used directly after unpacking, quantifying the emissions in the first hours/days is crucial if the data should also be used for exposure assessment for short-term exposure, especially if a peak-exposure is observed. For the flask method it is not possible to estimate peak exposure in the same manner compared to emission chambers. Thus, the correlation on emission rate (which is equivalent to the emitted concentration at constant area-specific air flow rate) is necessary in this case. We added a figure illustrating the linear correlation between area-specific emission rates (SERA values) for emission test chambers and microchambers at steady state in the revised manuscript (see 3.2., Figure 4).

The state of the art section does not present the problematic of the formaldehyde emission from toys and the existing methods to determine it, in a clear way. This section starts speaking about the limit of the indoor formaldehyde concentration of formaldehyde and then the emissions from wood-based panels is approached. The materials and regulations related to toys should be approached first and then the methods used to determine formaldehyde emissions from wood-based panels, one of the material that could be used in toys.

Response of the authors: The authors assume it as useful to introduce the health risks and the standard measurement methods for wood-based panels first. To introduce the problematic of the formaldehyde emission from toys without that background, we assume the introduction of the new amendment of the European toy safety directive as very difficult to understand without mentioning emission test chambers or EN 717-1 before. For these reasons, the authors decided to maintain the original logic structure of this section but processed the part on formaldehyde emission from toys to present it in a clearer way.

 The recent strict regulations of formaldehyde emission, as CARBII legislation or F**** and also the latest German legislation from Federal Ministry for the Environment, Nature Conservation and Nuclear Safety of Germany made to their testing method, that effectively lowers the formaldehyde emission levels of wood-based panels in Germany from the European emission level of 0.1 ppm (E1, EN 717-1) to 0.05 ppm were not treated. The standard method EN 16516 indicated in this legislation is only mentioned in the conclusions.

Response of the authors: The authors agree to the reviewer that a discussion of the new formaldehyde regulations might add important information to the manuscript. CARB (California Air Resources Board) Phase II sets different limits for formaldehyde emissions (0.05-0.11 ppm) depending on the sample type. In Germany, the national chemicals prohibition ordinance provides a concentration limit for formaldehyde of 0.1 ppm. In 2020 the analytical for formaldehyde for wooden building material in this regulation was changed from EN 717-1 to EN 16516 in order to provide a higher level of protection for the general public. A restriction process for formaldehyde is currently underway in Europe within the framework of the European chemicals regulation REACH. The procedure is currently not yet completed. European Chemicals Agency’s (ECHA’s) Risk Assessment Committee (RAC) recommends in its restriction proposal an emission limit value from objects of 50 μg/m³ (0.04 ppm), ECHA’s Socio-Economic Analysis Committee (SEAC) agreed to that opinion. But until now, there is no legislation in law on this subject. It is however true that the new standard method EN 16516 with an air change rate of 0.5 h-1, a higher humidity rate of 50% and a higher loading factor of 1.8 m2/m3) leads to a higher protection level compared to EN 717-1. Our room concentration estimations were also carried out considering an air change rate of 0.5 h-1(see equation (3)). This issue is now discussed in the introduction (Line 57-64 in unmarked version) and in the results part (Line 431-433 in unmarked version).

The example of emissions from carpets is somehow out of scope.

Response of the authors: We agree that carpets, as formaldehyde emission sources, are out of scope. But this study was one of the only one where emission chambers of different sizes where compared for formaldehyde emission measurements. We think it is relevant to mention it for that reason.

The sentence “Wood-based materials made of urea-formaldehyde resins are among the products that are responsible for the highest indoor formaldehyde emissions” is not really true, nowadays. The reference is from 1999. Nowadays, wood-based panels are produced with formaldehyde levels that complains strict regulations, even at solid wood level.

Response of the authors: The authors agree. This reference was deleted and replaced by more recent studies from 2018 measuring emissions from particleboards, without investigating their impact on the total indoor formaldehyde emissions (Line 49-50 in unmarked version).

The sentence “Often, the analysis of air samples is carried out after derivatisation with 2,4-dinitrophenylhydrazine (DNPH), which is provided in cartridges, and successive elution and liquid chromatography following ISO 16000-3” should be better contextualized, because this procedure is the analysis of formaldehyde in the air. It should be mentioned that normally the analysis of formaldehyde in the chamber method is carried using the photometric method. Formaldehyde is absorbed in water and the water is analysed photometrically using the acetylacetone method. Moreover, for DNPH the system is expensive (cartridges, solvents and columns) and the formaldehyde in the acetonitrile reagent should be checked on a regular basis. It is mentioned in the conclusions that “Additionally, other analytical techniques, such as photometry could be considered for air sample analysis to further reduce measurement costs”, but the photometric method is not addressed.

Response of the authors: Now we briefly discuss the differences between both analysis methods in the introduction (Line 63-66 in unmarked version). The authors consider both methods as suitable for formaldehyde analysis in combination with emission chamber testing. We used the DNPH method in our study because it was widely used and validated in our laboratory, contrary to the photometry method. We did not address the differences between the photometric and the DNPH methods experimentally because they are both validated and well established. We assume that it is out the scope of this work which focuses on emission test chamber and flask method comparisons.

I disagree that the flask method is widely used by official control laboratories (OCLs) in the EU member states to measure formaldehyde emission of products. In fact, the perforator and gas analysis methods are nowadays used by wood-based panels producers as quick methods for production control because the correlation to emission chamber testing is generally good. The accuracy and correlation of the flask method to the chamber methods are difficult to achieve due to small sample sizes, no air exchange, and high relative humidity (see DOI: 10.1039/c7ra06598a).

Response of the authors: To our best knowledge the terminus official control laboratories (OCLs) is commonly used for official authorities which enforce legislation and market control in the field of food, toys cosmetics and consumer products. OCLs are not responsible for wood-based panels producers and their quality control. None of the German OCLs uses gas analysis or perforator methods. We specified the meaning of “OCLs” (Line 74-76 in unmarked version). We completely agree that the accuracy of the flask method and its correlation with the chamber methods is poor. The suggested literature reference however only mentions the flask method very briefly and does not present experimental results to support this view, contrary to the cited reference from Maciej et al. (2011) (unfortunately only available in German). The flask method is still in practice used by several OCLs. The rapid alert system for consumer products Safety Gate of the European Commission contains some alarms related to toys and formaldehyde. Most of these alarms are based on analysis for formaldehyde related to the European toy standards and based on the flask method.

(https://ec.europa.eu/consumers/consumers_safety/safety_products/rapex/alerts/repository/content/pages/rapex/index_en.htm).

German OCLs are asking our institute, the BfR, to advise on reliable and cost-effective methods for formaldehyde emission measurements from toys. This topic was discussed with experts from different fieds in detail in BfR’s commission for consumer products and its sub-committees on analytics and toys. This work was conceived in this context, we detailed it in the manuscript (Line 87-90 in unmarked version). The Chemical and Veterinary Analytical Institute Münsterland-Emscher-Lippe (CVUA-MEL), one German OCL, was also directly included within this project and conducted the flask method measurements and provided real toy samples for our project, which were tested as positive for formaldehyde with the flask method which they used in routine for market control.

The observation “To our knowledge, microchambers have so far never been compared to large and regular emission chambers in terms of formaldehyde emission testing” is not truth. Please see the paper https://doi.org/10.1155/2018/4582383.

Response of the authors: We agree that a discussion of the paper by Hemmilä et al (2018) is useful. Therefore, this study was added and discussed in the manuscript. However, it is noteworthy that the microchamber used in that study (GP® Dynamic Microchamber from GP Georgia Pacific Chemicals, USA) is completely different in its dimension as the one we used (Micro-Chamber/Thermal Extractor™ (µCTE) from MARKES, UK) and has a much bigger volume (44 L instead of 44 mL). It is even bigger than our desiccator chambers of 24 L. The difference is now discussed in the manuscript (Line 113-116 in unmarked version).

In materials and methods, all the samples should be clearly described or a photo should be presented. The estimated area is not presented. A small description of the Micro-Chamber/Thermal Extractor™ (μ-CTE™) should be also included.

Response of the authors: Now the description of the samples contains more details (Table 1, Figure S1 and Table S1). A small description of the Micro-Chamber/Thermal Extractor™ (μ-CTE™) was included in the revised manuscript, together with a reference where more information is provided (Line 109-111 in unmarked version).

The dimensions of the particleboard panels for used in chamber method are too small. The units are probably not correct. In EN 717-1, the dimensions of test pieces are normally 0,5 m x 0,5 m x thickness, which should correspond to a total loading rate of 1 m²/m³. In case of dessicator method, the test pieces should have 150 mm x 50 mm and the total surface area of 1800 cm2 including ends, sides and faces.

Response of the authors: Indeed, the units for the two particleboard panels were not correct in the methods part, this has been corrected (Line 152 in unmarked version). The dimensions are 0.5 m x 0.5 m and 0.43 m x 0.5 m, which corresponds to a loading of 0.93 m2/m3. As we wanted to analyse the panels in both chambers and used panels already used in a previous study (Wilke et al. (2018)) for which we knew that formaldehyde would be emitted, we cut one of the boards: for this reason, the loading factor was not exactly 1 m2/m3. Anyhow, the goal of this study was not to strictly comply with the conditions of EN 717-1, but to apply real-life conditions for which the different emission test chambers were comparable. We insisted on temperature (23°C) and relative humidity (50 %) being equal while the sample area-specific air flow (ratio of air change rate and loading) were kept as similar as possible. For the desiccator method with the pieces, the loading depended on the available number of pieces. We put as much pieces as possible in the chamber to match the sample area-specific air flows of the microchambers.  More details on the sample dimensions are now given in Table 1 and Table S1.

The results are not always well explained. Why the results of the evolution SER along the time from puzzle pieces attained nearly a steady-state condition and this evolution is very different in case of particleboard samples?

Response of the authors: This a very important aspect of the discussion of our experimental results. It had only been discussed briefly in the first version of the manuscript and is now detailed (Line 309-316 in unmarked version).

In Figure 6, it is presented the calculated formaldehyde concentration for a 30 m³ room resulting from each puzzle sample Please explain why in line 360, it is mentioned “The sum of the puzzle plate and the number of associated pieces has been considered”.

Response of the authors: For realistic exposure assessment purposes, the influence of a whole puzzle set (plate and corresponding number of pieces, as a consumer would buy it) on the formaldehyde room concentration has to be considered. This is described now in more details (Line 409-410 in unmarked version). The number of pieces per puzzle set sample is presented in Table 1.

Considering this figure, the contribution of each puzzle piece for the indoor formaldehyde concentration in air for a 30 m³ room is very high in comparison with the indoor air guideline value (100 μg/m3).

Response of the authors: It is not the concentration for one puzzle piece, but for one complete puzzle set containing a plate and several pieces. This has been clarified in the manuscript (Line 409-410 in unmarked version).

Considering that the room (30 m3) is a CSTR (continuous stirred-tank reactor) with a flow of 15 m3/h, in which a formaldehyde source with 0.4 mg/(m2h), we will have a steady state concentration of C=0.4*A/15=0.026*A mg/m3. Given a concentration of more or less 0.004 mg/m3 in the room (see Fig. 6), the total surface of the sample should be around 0.15 m2, which is extremely large compared to a puzzle piece. Moreover, to exceed the WHO guideline, it will be necessary to have 3.75 m2 which corresponds to 1/3 of the total floor surface area of the room.

Response of the authors: The calculation is right, but the sample is a whole puzzle set so a surface of 0.15 m2 is plausible (See sample dimensions in Table 1). We agree that the concentration caused by one sample (max. 5 µg/m3) is small compared to the WHO guideline. A very large sample surface would be necessary to exceed the guideline. However, we think that the contribution of such samples should not be considered negligible, especially when the child plays in close proximity with the toy in a bad ventilated space with poor air change rate, and that further toy samples (we only analysed 8 samples, this is not representative of the market) may lead to even higher concentrations. Furthermore, to our opinion toys should not directly be tested against the indoor air guideline value for market surveillance, as also other and more prominent sources like building products or furniture contribute to the real indoor air concentration of formaldehyde. According to BfR’s opinion 005/2008, toys only should be allowed to contribute to 10 percent of the indoor air guideline value for indoor air (Line 424-426 in unmarked version).

(only available in German,  https://www.bfr.bund.de/cm/343/bfr_schlaegt_die_ueberpruefung_des_grenzwertes_der_din_norm_fuer_die_formaldehydausgasung_aus_holzspielzeug_vor.pdf )

Some minor revisions are also indicated in the manuscript.

Response of the authors: The suggested revisions were applied.

Reviewer 3 Report

The manuscript reports on the comparison of three sampling systems for measuring formaldehyde emissions from wooden toys.

Overall, the study is well presented and clear. The subject is highly relevant to scientists involved in the activity of controlling formaldehyde emissions. As of today, as the authors clearly say in the report, there is still a great need for information regarding the systems for sampling formaldehyde emissions and measure the concentrations of this hazardous compound in the environment. For this reason, in the opinion of this reviewer, it is highly recommended to publish the study presented by Even and coauthors. However, it is also strongly suggested to improve the text eliminating the flaws listed in the following.

Abstract

Lines 24-30. Please be quantitative and add data here. Specify what is meant by "similar emission profiles," "slightly higher levels," "equivalent results," "high variability." Some numbers will be much more informative and will eliminate the unnecessary vagueness in this paragraph.

Graphical abstract: It is maybe a little confusing. It is just a proposal made for improving the picture; can the authors eliminate the wooden toys' illustration? Or else move it outside of the light blue rectangle representing the exposure assessment? The object to be assayed is not a regular part of the sampling system.

P.4, line 130: BAM, can the authors clarify what BAM is?

Lines 172-173: Check whether the producer of this instrument is Agilent or Hewlett-Packard.

Equation 3. Can the authors explain in more detail why a Cindoor assessment is needed?

Equation 4: please explain what is the use and meaning of the Offset.

Section 2.4

Please explain which outlier test was performed, clarifying the null and alternative hypotheses, the significance, and the power of the test used. Report also what p-values were computed.

Line 216. Please clarify what is the acceptance range and based on what criteria it was established.

Equations 5 and 6. The Q1 and Q3 quarters have not been defined. It is not clear what they mean.

Figures 1, 2, 3, 4, and S3. Please clarify what is the error bar meaning in the figures. Is this mean SER ± standard deviation or mean SER ± confidence interval semiamplitude?

Lines 323-324. Data do not support the authors' statement. Please provide data, a table, or a graph form that the readers can understand the absence of correlation observed.

Lines 331-332. Please provide the data and the statistical analysis, which led to this conclusion (no significant difference between the values).

Lines 351-355. The data in figure 5a do not strongly support the statement)- In this figure, no error bars are reported. What were the errors of the measurements for pieces and plates of sample #8?

Tables 3 and S1. The authors should report these results as mean value ± standard deviation or another estimate of the uncertainty affecting the Croom data's measurement (or computation).

Author Response

The authors would like to thank the reviewer for the constructive and very detailed comments that helped us to improve the quality of our manuscript. Please find below our point by point responses to all comments:

The manuscript reports on the comparison of three sampling systems for measuring formaldehyde emissions from wooden toys.

Overall, the study is well presented and clear. The subject is highly relevant to scientists involved in the activity of controlling formaldehyde emissions. As of today, as the authors clearly say in the report, there is still a great need for information regarding the systems for sampling formaldehyde emissions and measure the concentrations of this hazardous compound in the environment. For this reason, in the opinion of this reviewer, it is highly recommended to publish the study presented by Even and coauthors. However, it is also strongly suggested to improve the text eliminating the flaws listed in the following.

Abstract

Lines 24-30. Please be quantitative and add data here. Specify what is meant by "similar emission profiles," "slightly higher levels," "equivalent results," "high variability." Some numbers will be much more informative and will eliminate the unnecessary vagueness in this paragraph.

Response of the authors: Concrete data were added to the abstract (Line 24,25,28 in unmarked version).

Graphical abstract: It is maybe a little confusing. It is just a proposal made for improving the picture; can the authors eliminate the wooden toys' illustration? Or else move it outside of the light blue rectangle representing the exposure assessment? The object to be assayed is not a regular part of the sampling system.

Response of the authors: We modified the graphical abstract and moved the wooden toys’ illustration outside of the blue rectangle.

P.4, line 130: BAM, can the authors clarify what BAM is?

Response of the authors: This has been clarified (Line 148 in unmarked version).

Lines 172-173: Check whether the producer of this instrument is Agilent or Hewlett-Packard.

Response of the authors: Agilent was created in 1999 as a spin-off of Hewlett-Packard. The device is from 1997 and therefore an HP device. The address of Waldbronn, Germany is given because it is the European headquarter of Agilent Technologies.

Equation 3. Can the authors explain in more detail why a Cindoor assessment is needed?

Response of the authors: Evaluating Cindoor allows a direct comparison with the indoor air guideline and therefore a reliable risk assessment. This was added in the manuscript (Line 411-412 in unmarked version).

Equation 4: please explain what is the use and meaning of the Offset.

The use of the offset allows a direct comparison of the differences between emission test chambers for different samples, this was now added to the manuscript (Line 227-228 in unmarked version). 

Section 2.4

Please explain which outlier test was performed, clarifying the null and alternative hypotheses, the significance, and the power of the test used. Report also what p-values were computed.

Line 216. Please clarify what is the acceptance range and based on what criteria it was established.

Equations 5 and 6. The Q1 and Q3 quarters have not been defined. It is not clear what they mean.

Lines 323-324. Data do not support the authors' statement. Please provide data, a table, or a graph form that the readers can understand the absence of correlation observed.

Response of the authors: The analysis of the ratio between flask method values and emission rates was replaced by the evaluation of the linear correlation between both terms (similar to Hemmilä et al. (2018)), which appears more plausible and straight forward to us. The statistical parameters (R2 and p-value) and acceptance range are introduced in 2.3 (Line 229-232 in unmarked version).

Figures 1, 2, 3, 4, and S3. Please clarify what is the error bar meaning in the figures. Is this mean SER ± standard deviation or mean SER ± confidence interval semiamplitude?

Response of the authors: The error bars correspond to the standard deviation. This was clarified in the manuscript.

Lines 331-332. Please provide the data and the statistical analysis, which led to this conclusion (no significant difference between the values).

Response of the authors: We now provided the relative standard deviation when the sample number n was ≥ 2. For the other samples, the measurement uncertainty was evaluated to be ±30% (see Table S2). Indeed, it is not always possible for the OCLs to systematically repeat tests as the samples are taken from the market and their quantity is limited. Moreover, we considered only the pieces which had a similar color as the one we put in the microchambers as lacquers may have an influence on the formaldehyde emissions. Our main concern was to get an idea of whether and how the flask method can predict real emission values for real toy samples. For an extensive validation of the flask method against the test chambers, a different experimental approach would be necessary. A homogeneous material with the same shape as toy samples would be useful.

Lines 351-355. The data in figure 5a do not strongly support the statement)- In this figure, no error bars are reported. What were the errors of the measurements for pieces and plates of sample #8?

Response of the authors: For us it is unclear why Figure 5a (now Figure 6a) should not support the statement. The emission rates from the plates (dashed lines) are higher for #5 and #7 than the ones from the pieces, for #8 it is the contrary. We were not able to repeat the desiccator experiments due to a lack of identical sample sets and the need for maximal loading for test chamber comparison (see also Figure 3). Therefore, we were not able to report error bars for the desiccator experiments. For microchamber experiments, two pieces of each samples were studied. “n=1” was added in the legends of Figure 5 and Figure 6 (now Figure 6 and 7).

Tables 3 and S1. The authors should report these results as mean value ± standard deviation or another estimate of the uncertainty affecting the Croom data's measurement (or computation).

Response of the authors: The standard deviation was added to Table 3 and Table S2 (when available for Table S2: the flask method measurements were not always repeated as many sample pieces are placed together in the flask).

Round 2

Reviewer 1 Report

In this form, the paper deserves to be published in this journal.

This manuscript is a resubmission of an earlier submission. The following is a list of the peer review reports and author responses from that submission.